# Identifying Risk Indicators for Natural Hazard-Related Power Outages as a Component of Risk Assessment: An Analysis Using Power Outage Data from Hurricane Irma

**Sang-Guk Yum** [1] , **Kiyoung Son** [2] **, Seunghyun Son** [3] **and Ji-Myong Kim** [4,*]

[1]  Department of Civil Engineering and Engineering Mechanics, Columbia University, New York, NY 10027, USA; sy2509@columbia.edu
[2]  School of Architectural Engineering, Ulsan University, Ulsan 44610, Korea; sky9852111@ulsan.ac.kr
[3]  Department of Architectural Engineering, Kyung Hee University, Suwon 17104, Korea; seunghyun@khu.ac.kr
[4]  Department of Architectural Engineering, Mokpo National University, Mokpo 58554, Korea
*   Correspondence: jimy6180@gmail.com

**Abstract:** Extensive use has been made of lifecycle-cost assessment to enhance the cost-effectiveness and resilience of facilities management. However, if such assessments are to be truly effective, supplemental information will be needed on the major costs to be expected over buildings' entire lives. Electricity generation and distribution systems, for example, are absolutely indispensable to industry and human society, not least in the operation of buildings and other infrastructure as networks. The widespread disruption that ensues when such power systems are damaged often carries considerable repair costs. Natural disasters likewise can cause extensive societal, economic, and environmental damage. Such damage is often associated with lengthy power outages that, as well as being directly harmful, can hinder emergency response and recovery. Accordingly, the present study investigated the correlations of natural hazard indicators such as wind speed and rainfall, along with environmental data regarding the power failure in Florida caused by Hurricane Irma in 2017 utilizing multiple regression analysis. The environmental data in question, selected on the basis of a thorough literature review, was tree density. Our analysis indicated that the independent variables, maximum wind speed, total rainfall, and tree density, were all significantly correlated with the dependent variable, power failure. Among these, rainfall was the least significant. Despite there being only three independent variables in the model, its adjusted coefficient of determination (0.512) indicated its effectiveness as a predictor of the power outages caused by Hurricane Irma. As such, our results can serve the construction industry's establishment of advanced safety guidelines and structural designs power transmission systems in regions at risk of hurricanes and typhoons. Additionally, insurance companies' loss-assessment modeling for power-system facilities would benefit from incorporating the three identified risk indicators. Finally, our findings can serve as a useful reference to policymakers tasked with mitigating power outages' effects on infrastructure in hurricane-prone areas. It is hoped that this work will be extended, facilitating infrastructure restoration planning and making societies and economies more sustainable.

**Keywords:** natural hazard; risk management; power system failure; disaster management

## 1. Introduction

Hurricanes, typhoons, and tropical cyclones have been occurring more frequently and increasing in intensity because of global warming [1–4]. Climate change also appears to affect the tracks of these

extreme weather events, and thus the amount of damage they cause via high winds and flooding [5]. According to Kreimer and Amond [6] and Chang [7], this has increased both the direct and indirect costs of natural-disaster damage as a whole. A Munich Re report [8] revealed that the 2017 damage caused to the United States by hurricanes—around $220 billion—was the highest ever recorded, due to the trio of Hurricanes Harvey, Irma, and Maria. Such events can cause severe economic, environmental, and societal damage, not least through lengthy interruptions to the supply of electric power, one of the most important of which is increases to the lifecycle costs of buildings and other infrastructure. In the U.S., the costs of damage specifically due to power outages have been increasing [9,10], and are closely associated with hurricanes. Hurricane Irene, for example, deprived 6.5 million people of power in 2011, while the parallel figure for Hurricane Sandy the following year was 8.5 million [9].

Many studies have investigated the increased damage caused by power outages linked to natural hazards and the ensuing social and economic problems [10–13]. Other research has emphasized that, because the severity of natural hazards is likely to increase in the future, making power systems resilient to such hazards will only become more difficult [14–16]. Some recent work on hurricane damage has focused not only on direct losses, but also on the enormous cost of restoration, which was $150 billion in the U.S. between January 2004 and December 2005 [17]. Hurricane damage is an inescapable consequence of hurricanes striking built-up areas. Nevertheless, the creation of a prediction model for such damage can form an important part of planning for, and effective responses to, such events. Electric power failure has multiple socio-economic impacts, and can cause serious problems for hospitals, schools, and other critical public infrastructure. However, if utility companies and governments can understand the critical correlations between natural hazard variables and power outages, and design their emergency-response and mitigation plans accordingly, their localities' ability to respond to natural hazards will be greatly improved. Therefore, using multiple regression analysis of a U.S. electric power company's outage data from Hurricane Irma, the present study seeks to identify which hurricane variables are most closely related to power outages. This is a necessary first step toward the better emergency-response plans and prompt restoration of the power facilities mentioned above.

## 1.1. Research Background

Lifecycle assessment studies are frequently conducted, often focusing on buildings' design stages as a means of making their management more cost-effective. Sometimes, they look at facility management as an aspect of asset management. However, most lifecycle-cost studies pay relatively little attention to buildings' operation and management stages. Though natural hazards remain very difficult to predict, the frequency of severe ones has increased [18]. Moreover, Hurricanes Katrina in 2005, Ike in 2008, and Sandy in 2012 caused damage worth $108 billion, $29.5 billion, and $71.4 billion, respectively [19,20]. It is well known that power outages have significant economic impacts on the operation of facilities essential to society. The negative effects of significant damage to power-infrastructure systems induced by hurricanes and other storms proliferate outward to households, healthcare providers, schools, government facilities, and whole communities, economies, and societies [21]. Protection of these vulnerable lifelines, or—where protection is impossible—their rapid restoration, is essential to reducing risks to life and property [22]. Therefore, assessment of the likely impact of extreme natural hazards would seem critical to effective assessment of the costs of operating and managing buildings and infrastructure over their entire lifecycles, as well as to decision-making for utility companies and emergency managers.

By the same token, blackouts are an important metric of hurricanes' impacts on particular areas. The present research addresses hurricane-related power outages' importance to the operation and management stage of building and infrastructure system lifecycles from a sustainability viewpoint, utilizing actual power-failure data from when Hurricane Irma struck Florida.

*1.2. Research Objective*

The principal goal of this research is to assess the significance of specific hurricane and power outage variables through multiple regression modeling of historical damage data. First, based on a review of the relevant literature, it will attempt to capture the major expected factors in hurricane-related power. Second, it will use multiple-regression analysis to reveal the relationship between 1326 power outages (the dependent variable) and various characteristics of Hurricane Irma and other aspects of the natural environment (the independent variables). Specifically, this power outage data consists of relative outage frequency (ROF), meaning the number of power outage events per 0.1 million people. The natural hazard indicators include wind speed and rainfall. Tree density will also be used, since the immediate cause of power outages during natural disasters is often fallen trees [23].

## 2. Literature Review

According to Sissine [24], population growth and rapid urbanization have rendered high-performance buildings an important possible solution for sustainability. If the need for such buildings is to be met, however, their long-term environmental and economic sustainability should be considered simultaneously [25]. The U.S. Green Building Council [26] has proposed that the three main sustainability factors for quality of life are the present and future harmony of society, the economy, and the environment. Thus, balanced long-term pursuit of sustainable development should take account of all three of these factors across buildings' entire lifecycles. To date, however, scholarship on buildings' sustainability performance has focused on short-term social impacts such as fatalities and short-term economic ones such as repair costs, while largely ignoring long-term impacts, especially on the environment [27–30]. Wei et al. [31] suggested that environmental factors have received less attention than economic and social ones due to lack of consensus about what environmental criteria are important, and how they should be defined and measured. Among those studies that have investigated the environmental impacts of natural disasters that strike buildings and other built infrastructure, Chang and Shinozuka's [32] work on seismic risk is perhaps especially important. It yielded an expanded lifecycle-cost framework using a hypothetical water delivery system, with performance-level definitions and criteria for minimizing the lifecycle costs related to the repair of pipeline systems after earthquakes. Shinozuka et al. [33] subsequently proposed a lifecycle-cost estimation framework for bridges at risk of being struck by earthquakes, which includes not only initial bridge construction costs, but also damage repair costs. Similarly, Wei et al. [34] proposed that the estimation of buildings' true lifecycle costs for the purposes of sustainable development should include the potential costs associated with seismic events. Post-event recovery's impacts on systems' lifecycles has also recently been emphasized, with such discussions covering energy demand, building performance metrics, and sustainability rating systems [31,35–40].

According to Sinisuka and Nugraha [41], power generation's role in lifecycle cost includes both deterministic costs, such as asset acquisition and operation/management costs, and probabilistic ones, such as the costs of failures, repairs, and gross-loss margins. The same study reported that probabilistic costs were mainly associated with power systems' maintenance. Power outages related to natural hazards have also been recognized as a huge societal problem, in the first instance via research on earthquakes. For instance, Shinozuka et al. [33,42,43] led an investigation into power outages caused by earthquakes for the U.S.'s Multidisciplinary Center for Earthquake Engineering Research, mainly focusing on the restoration of power in the wake of extreme events. Power systems' fragility curves have also been investigated as a means of assessing damage from power outages before such events have taken place [44,45]. According to Guikema and Nateghi [46], many electric power companies use the Outage Management System (OMS) to record power outages, and historical data gathered from this system have contributed to effective prediction of the magnitude of power outages caused by natural hazards such as hurricanes. However, it remains difficult to forecast power outages using OMS, since extreme climatological events caused by global warming are more likely to happen in the future than they were in the past.

Hurricane-induced power outages cause considerable direct and indirect damage both to utility companies and to end-users of electricity, and thus impose significant restoration and recovery costs on whole affected regions [23]. They are caused by several factors, most notably high winds and flooding [46]. However, most of the relevant studies have focused on direct wind-induced damage to elements of power systems such as poles and power lines, as well as indirect wind damage, e.g., similar damage caused by fallen trees. Davison et al. [47], for example, concluded that wind gusts were correlated with power outages in a certain region, and that such damage could be mitigated by different types of vegetation cover; however, they did not consider other variables such as flooding and precipitation. Liu et al. [22] extended Davison et al.'s [47] work, but only by adding information such as the duration of the maximum wind speed and soil status. Additionally, Reed [48] made novel use of Geographic Information System data to study major storms in the northwestern U.S., but again concluded that storm-induced wind speed was the most impactful factor for power outages there. Nevertheless, Hurricane Sandy caused huge power outages in Manhattan due to inundation of the power system [46], and such possibilities deserve more scholarly attention than they have been given.

Among studies not narrowly focused on wind, Han et al. [49] introduced hurricane intensity and size as factors in power outages' spatial distribution on the U.S. Gulf Coast. The same study also linked physical damage to power systems to the time that elapsed between one hurricane and another. Guikema et al. [50] and Nateghi et al. [51] developed a model for predicting power outages prior to hurricanes, with more accuracy than previous studies of the same type that had relied solely on publicly available data. Even though such prior studies' datasets had been large, they arguably did not take account of wide enough arrays of natural hazard indicators. Balijepalli et al. [52] developed a promising method of estimating power outages related to lightning storms, using a bootstrap method and Monte Carlo simulations. Quiring et al. [53], on the other hand, looked at seldom-studied power outage variables such as soil moisture. Even though they concluded that soil did not have a significant impact on outages, their work suggested that vegetation cover might be a useful indicator of soil stability, and thus of how likely power poles are to fall.

Very recently, lifecycle-cost assessments that take account of natural hazards have become more common. However, most such studies continue to include only the costs of repairing damage in the immediate aftermath of natural disasters [54]. Due to global warming and rapid urbanization, the frequency and intensity of such disasters are both increasing [25]. To supplement and support the findings of the prior studies reviewed above, therefore, new, more holistic approaches will be needed. The current study helps to fill that gap.

## 3. Research Methods

### 3.1. Case Study Approach

Irma, a category 5 hurricane on the Saffir–Simpson Scale, devastated the southeastern part of North America, especially Florida, between 30 August and 14 September 2017 (Figure 1). In all, the storm caused $77.16 billion in property damage and more than 100 fatalities. Its 10 min maximum sustained wind speed was 209 kph. The present research targeted an area in Florida covered by one of that state's largest electric system companies, which operated 18 centers, serving 35 counties. The recorded ROF data were limited to the dates in early September during which Hurricane Irma passed through that area.

### 3.2. Dependent Variable

As the dependent variable for multiple regression analysis, this study used data on 825 ROF that were received by the target system's 18 operations centers during Hurricane Irma's landfall in Florida.

### 3.3. Independent Variables

Meteorological wind parameters such as maximum wind speed, forward wind motion speed, and radius have all been utilized as key indicators of hurricane damage levels [55,56]. However, according to Burton [55], Watson and Johnson [57], and Vickery et al. [58], maximum wind speed is the best of these. In the current study, therefore, it was used as the independent variable relating to wind. However, following Choi and Fisher's [59] and Brody et al.'s [60] proposals that rainfall could be the dominant indicator of the extent of damage from hurricanes, total rainfall during Hurricane Irma was also adopted as an independent variables.

Aspects of the built environment can also serve as useful indicators of regions' vulnerability to hurricanes, and for estimating their damage and losses [61]. Mitchell [62] reported that tree falls often led to power line failures, especially when wind loading and soil movement caused by hurricanes caused whole trees to be uprooted. However, fallen tree branches can also cause power system failures [63]. In this study, therefore, tree-density data for the target area were collected from the U.S. Federal Government. These data recorded the number of trees per 1000 m$^2$ in the areas where power outages were reported to the local utility company, with trees being defined for this purpose as woody plants greater than 2.5 cm in diameter and extending more than 1.38 m above the ground surface.

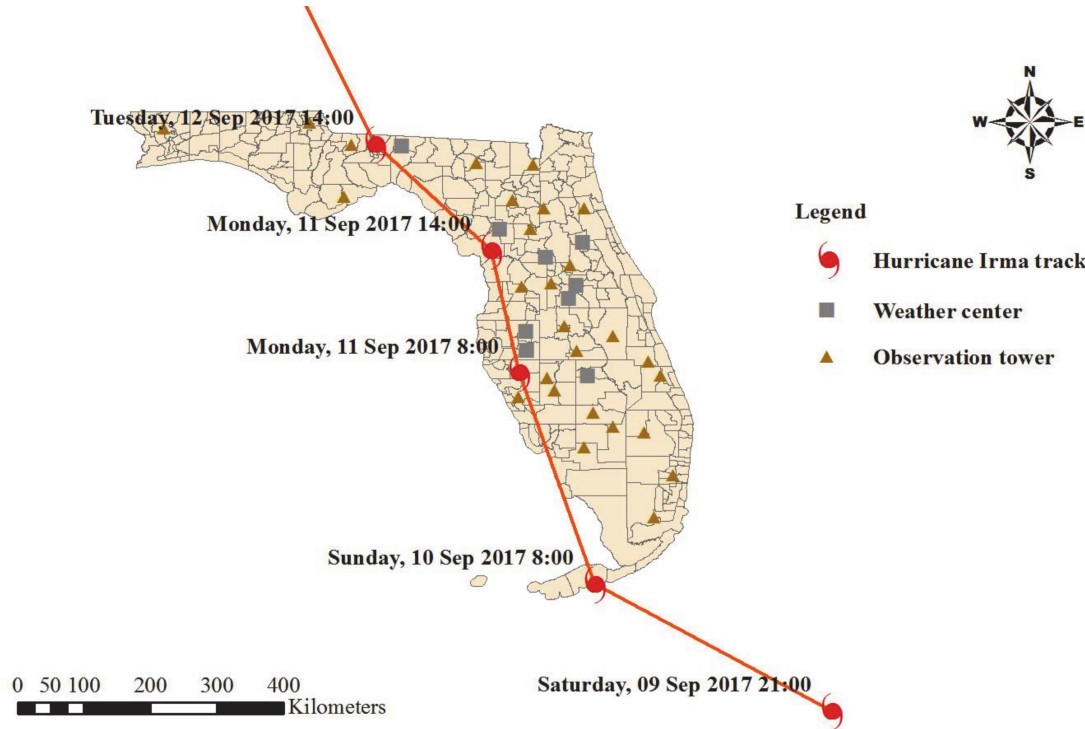

**Figure 1.** Track of Hurricane Irma, with locations of weather centers and observation towers.

### 3.4. Data Collection and Management

Table 1 presents further details of the present study's dependent and independent variables. Data on the regional hurricane-induced power outages caused by Hurricane Irma in 2017 were collected and used as the dependent variable. The focal utility company, which provided that data, served approximately 3.1 million customers in the north central, north coastal, south central, and south coastal areas of Florida. The outage data were collected from all 18 power company operations centers within 12 counties in the target area. The 10 min maximum wind speed and total rainfall data were collected from nine weather centers located close to those operations centers. Those weather centers monitor hurricanes and their meteorological characteristics, i.e., wind speed, movement, rainfall, pressure, radius, and intensity. Information about each city with an operations center and/or weather center is

shown in Table 1. The numbers by county of reported power outage incidents during Hurricane Irma are shown in Table 2.

**Table 1.** Data types and sources.

| Variable | Description | Unit | Data Source |
|---|---|---|---|
| ROF | Number of power outages per 100,000 people | Number of power outages/100,000 | Power company operations centers ($n = 18$) (Apopka, Deland, Jamestown, Longwood, Inverness, Monticello, Ocala, Buena Vista, Clermont, Se Orlando, Highlands, Lake Wales, Winter Garden, Clearwater, Seven Springs, St. Petersburg, Walsingham, Zephyrhills Bay) |
| Maximum sustained wind speed | 10 min maximum sustained wind speed (based on the weather station closest to the point of the outage) | m/s | Weather stations ($n = 9$) (Dover, Balm, Apopka, Bronson, Ocklawaha, Sebring, Monticello, Pierson, Avalon) |
| Total rainfall | Total amount of rainfall per day | cm | |
| Tree density | Number of trees per 1000 m$^2$ where power outage data were reported | Number of trees/1000 m$^2$ | Florida Geographic Data Library |

**Table 2.** Reported number of power outages from Hurricane Irma per county.

| County | N |
|---|---|
| Lake | 163 |
| Jefferson | 120 |
| Wakulla | 86 |
| Polk | 82 |
| Citrus | 78 |
| Pinellas | 78 |
| Highlands | 75 |
| Walton | 62 |
| Seminole | 42 |
| Orange | 39 |
| Total | 825 |

The raw power outage data initially comprised more than 1500 incidents. However, the focal weather centers missed some of the meteorological data pertaining to Hurricane Irma, due to being unable to cope with the speed of its unexpected wind gusts. Therefore, only 825 power outage incidents from the power company's dataset had weather center meteorological data corresponding to them. Thus, only those 825 incidents were retained for further analysis. Tree-density information, pertaining specifically to the areas where those same 825 reported power outages occurred, was provided by the Florida Geographic Data Library.

*3.5. Multiple Linear Regression Analysis*

Regression analysis is a statistical method for predicting the tendencies of collected data. Multiple regression analysis is used to show the linear relationship between more than two independent variables and a dependent variable. The adjusted coefficient of determination ($R^2$) is used to establish the variances and correlations among the independent variables selected for use in the linear regression

model. In addition, $R^2$ plays the important role of checking the multiple regression models' goodness of fit; and the total variation in the dependent variable (i.e., ROF in this case) is explained as a proportion of the value of $R^2$ regarding the variability of the independent variables (i.e., here, maximum wind speed, total rainfall, and tree density).

As such, $R^2$ can be interpreted as approximately how well the linear prediction model can explain the real data. It ranges from 0 to 1, and the closer it is to 1, the better the model's predictive power. It can be estimated by the sum of squares of residual and regression, and the total sum of squares. Those values are also used for estimating *F* in the analysis. $R^2$ has a tendency to increase as the numbers of independent variables are increased. To improve this inherent drawback of the parameter, adjusted $R^2$ is usually used for multiple linear regression analysis. Additionally, analysis of variance (ANOVA) is used for confirming the significance of the regression model. When the significance value produced by an ANOVA is less than 0.05, the regression model is regarded as significant.

Additionally, the present study conducted a normality test for its regression analysis. Such a test can be used for validating that the residuals of the dependent variable are normally distributed. In practice, a *p*-value larger than 0.000 indicates that the residuals of the variable are normally distributed. In the present study, the normality test was conducted to check the normality of the ROF data before multiple regression analysis was conducted. After verifying that the dependent variable was normally distributed, multiple regression analysis investigated the relationship between ROF and the independent variables. That regression model yielded the predicted trend of the analyzed data, as shown in Equation (1). The straight linear relationship of the function indicates the relationship between the three independent variables and the dependent variable. Specifically, the equation shows that the ROF can be determined along with the independent variables (i.e., $X_1$, $X_2$, and $X_3$) and the regression coefficients (i.e., $\alpha$, $\beta$, $\gamma$, and $\omega$) of each independent variable. Thus,

$$ROF = \alpha + \beta^* X_1 + \gamma^* X_2 + \omega^* X_3 \tag{1}$$

where $\alpha$ is a constant; $\beta$ is the slope of tree density; $\gamma$ is the slope of total rainfall, and $\omega$ is the slope of maximum wind speed.

From the multiple linear regression analysis results, one can estimate the variance of inflation factor (VIF). VIF can be used as an indicator of multicollinearity among the independent variables. From VIF, the degree of correlations among independent variables can be determined, with larger figures indicating that an independent variable may depend on other independent variables: indicating that it is not, in fact, meaningfully independent. Therefore, analyses that include independent variables with large VIF values can have biased and unpredictable results; and specifically, when VIF is greater than 10, the researcher should consider removing it from the model. A beta (standardized) coefficient is applied to determine the degree of impact of independent variables on the dependent variable in multiple regression analysis, with larger beta coefficients indicating that such impacts are stronger.

## 4. Results

Table 3 presents descriptive statistics for the dependent variable and independent variables used in this study's model. N stands for the total number of ROF data points during Hurricane Irma's passage through Florida in September 2017.



**Table 3.** Descriptive statistics of variables.

| Category | Reported Cases | Mean | SD |
|---|---|---|---|
| **Dependent Variable** | | | |
| ROF | 825 | 1.955 | 2.001 |
| **Independent Variables** | | | |
| Maximum sustained wind speed (m/s) | 825 | 15.563 | 12.821 |
| Total rainfall (cm) | 825 | 0.291 | 1.051 |
| Tree density (number of trees/1000 m$^2$) | 825 | 3.134 | 1.432 |

Being larger than 0.05, our normality test's significance level of 0.24 indicated that the dependent variables were normally distributed. The multiple regression analysis results are shown in Figures 2 and 3. The *P–P* plot and histogram of the standardized residual (Figure 2) indicate that the residuals in the regression model were normally distributed, while the scatter plot (Figure 3) indicates that the residual's variance was both constant and randomly distributed, i.e., homoscedastic.

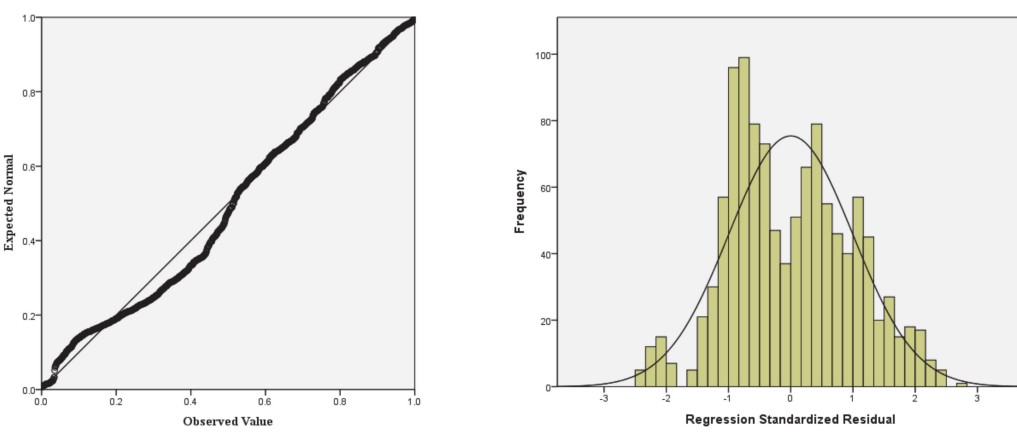

**Figure 2.** *P–P* plot and histogram of standardized residual from regression analysis.

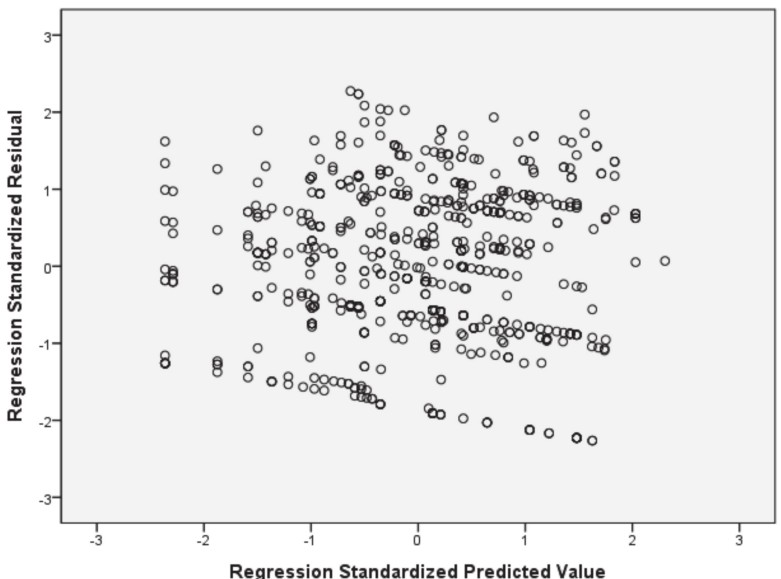

**Figure 3.** Scatter residual plot from regression analysis.

Analysis of variance (ANOVA) confirmed the above findings. When ANOVA yields a significance level smaller than 0.05, it indicates that the regression model is significant. In this case, it was 0.00,

showing that the sampled power outages were related to the independent variables linearly. As shown in Table 4, adjusted $R^2$ was 0.512, and the value of *F* was very high (202.768). This meant that 51.2% of the variance in the dependent variable could be explained by all of the independent variables collectively.

**Table 4.** Summary of ANOVA and multiple regression modeling.

| | Sum of Squares | df | Mean Square | F | Sig. | Adjusted $R^2$ |
|---|---|---|---|---|---|---|
| Regression | 454.200 | 5 | 90.84 | 202.768 | 0.000 | 0.512 |
| Residual | 411.712 | 919 | 0.448 | | | |
| Total | 895.912 | 924 | | | | |
| | (2) | | | | | |

The regression analysis results presented in Table 4 were then checked for statistical relations among the variables. This revealed that maximum sustained wind speed, total rainfall, and tree density were all significant factors, at a confidence interval (CI) of 95% ($p < 0.05$). Additionally, all VIF values were below 10, meaning that there was no multicollinearity among them. The standardized coefficients in the regression analysis indicate the relative strengths of the independent variables' impact on the dependent variable. Here, as shown in Table 5, tree density (0.211) had the largest such impact, followed by rainfall (0.118) and wind speed (0.092).

**Table 5.** Coefficients of multiple regression analysis.

| Variable | Non-Standardized Coefficient | Standardized Coefficient | Significance Probability (*p*-Value) | Collinearity Statistics (VIF) |
|---|---|---|---|---|
| (Constant) | 0.129 | 0.259 | | |
| Maximum sustained wind speed (m/s) | 0.112 | 0.092 | 0.024 * | 1.283 |
| Total rainfall (cm) | 0.227 | 0.118 | 0.001 * | 1.309 |
| Tree density (number of tree/1000 m$^2$) | 0.094 | 0.211 | 0.025 * | 1.016 |

Note. * $p < 0.05$; VIF = variance of inflation factor.

## 5. Discussion

Natural disasters such as landslides, earthquakes, floods, hurricanes, and wildfires are serious threats to sustainable development. As well as direct losses to the natural capital, they have profound impacts on whole countries' economic growth and sustainability [64]. In part, this is because they can cause or exacerbate imbalances in the supply and demand of social resources [65]. According to Fang et al. [64], due to recent more frequent and intense natural disasters, sustainable development has become a more urgent priority for many societies. Therefore, advanced decision-making processes in the context of natural hazards is increasingly being recognized as critically important.

To achieve more robust estimation of whether construction projects are economically sustainable, scholars have recommended that risk modeling incorporate additional factors such as natural disasters and special vulnerabilities of the local built environment, rather than continuing to focus narrowly on accidents at construction sites [66,67]. Sustainable risk management also needs plentiful information on potential risks if they are to be effectively managed. Especially in the construction industry, such management needs to take more account of risks to the environment, society, and the wider economy than it currently does. Therefore, sustainable construction should focus on balanced development across social, economic, and environmental needs [68,69]. Within the field of sustainable risk management more generally, the most-used method is the management of potential risk factors and uncertainty, which are inherently difficult to predict [70]. For effective decision-making about natural hazards, identifying risk indicators for those hazards is a key first step.

The present study's proposed damage prediction model utilized natural hazard indicators from when Hurricane Irma struck Florida. Its multiple regression analysis showed that the meteorological variables and one topographical variable were all significantly correlated with power outages during Irma. Our normality test's significance level of 0.24 showed that the dependent variable was normally distributed, while our ANOVA's significance level of 0.00 indicates that the selected independent variables had a linear relationship with the dependent one. Together, based on the adjusted $R^2$ of 0.512, these natural hazard variables explained 51.2% of the variation in the sampled power outages. The other 48.8% of such variation was not explained by the selected variables. Furthermore, the *p*-values of the coefficients of the independent variables in the regression model all indicate significance. Thus, the three selected independent variables—wind speed, rainfall, and tree density—can be used effectively in the prediction of power outages caused by natural hazards such as hurricanes.

These findings support those of various previous studies mentioned in the foregoing literature review. Specifically, wind speed's statistically significant relation to power outages supports the prior conclusions of Burton [55], Watson and Johnson [57], and Vickery et al. [58]. Similarly, the meaningful relation between rainfall and power outages identified in the present study supports previous findings by Choi and Fisher [59] and Brody et al. [60]. Lastly, tree density was revealed to be significantly correlated with power outages during hurricanes, supporting prior results by Mitchell [62] regarding tree- and tree-branch-related power outages.

Additionally, the regression model in Equation (1) explained the straight-line relationship between the dependent and independent variables in the present study. The non-standardized regression coefficients of the independent variables, 0.112, 0.227, and 0.094, were utilized for estimating the trend of ROF corresponding to these natural hazard indicators. Using the regression model and the coefficients, the present study revealed the following. First, when the maximum wind speed increased by 1 m/s, the ROF increased by 11.2%. Second, when total rainfall increased by 1 cm, the ROF increased by 22.7%. And third, when tree density increased by 1 tree/1000 $m^2$, the ROF increased by 9.4%. The total number of ROF was 3129 when Hurricane Irma made landfall in Florida, and our regression analysis estimated such ROF as 3284, i.e., within a ±95% interval of the actual ROF. As such, these non-standardized coefficients can be used effectively for estimating ROF. Since the actual cost of damage from the power outages caused by Hurricane Irma was not utilized in the present study, due to lack of information (an important limitation), the similarity of the estimated and actual ROFs mean that even more accurate estimation may be possible in future research.

The above findings are also expected to help power system companies improve their preparation for extreme natural hazards in their service areas, including via the adoption of underground cable systems where appropriate. Our results also afford an opportunity to improve traditional lifecycle-cost estimation, by shifting its focus away from the design phase and onto operation and management costs—not only of electric power systems, but also of buildings and coastal infrastructure, in areas prone to extreme weather events.

Insurance companies and construction companies will likely find it advantageous to modify their business plans and construction techniques to reflect natural hazards' impacts on their risk exposure, maximum loss, and so forth. Specifically, in the construction industry, the findings of the present study can be used to help estimate potential losses from power outages by line of business, i.e., commercial, residential, and industrial. Based on estimated damage cost, constructors would be able to give reasoned consideration to the use the underground power system networks as a means of reducing losses from power outages caused by hurricanes (among other, more rational methods of construction). The insurance industry, for its part, can benefit from using the natural hazard indicators we identified, for risk-management and risk-mitigation of insured properties or infrastructure facilities, and thus maximizing their profits. Furthermore, policymakers can refer to the results of the present research when, for example, forecasting the risk factors in hurricane-prone regions and estimating direct and indirect losses to infrastructure facilities and commercial properties. This information, in turn, could be utilized for establishing effective restoration plans that mitigate business interruption;

and specifically, tree-related power outages could be mitigated through more careful government-led management of vegetation distribution and tree species in hurricane-prone areas. In the long term, our findings regarding power outage risks can assist advanced risk assessment responses to climate change, by facilitating better-informed asset-management decision making.

Power transmission systems are vulnerable to extreme wind hazards, and the prompt restoration of such systems is critical to industries, utility systems, households, hospitals, and an array of other components of complex societies. Hurricane-induced power outages cause considerable direct damage such as repair costs, and significant indirect ones such as business interruption, in the impacted regions. A number of previous studies have emphasized the long-term repair costs, in time as well as money, of restoring power systems in the wake of disasters [71]. While restoration of power facilities is important, however, it is also very valuable to predict the damage that natural hazards are likely to cause before they occur. Developing vulnerability curves for natural disaster indicators caused by extreme events may be useful to estimating potential damage, as well as the places where power systems are most vulnerable to hurricanes. The natural hazard variables for power failure that we identified above, as well as others that may yet be identified, will be vital to the effective development of such curves.

The findings of the present study can also usefully be extended to the quantification of risk indicators for electric power transmission facilities construction, including socio-economic vulnerabilities arising from power outages, and not just those outages per se. Moreover, our findings regarding some key risk indicators for power failure can contribute to reducing operation and management costs, which are the greatest component of lifecycle cost [41]. Better monitoring of the facilities that may be vulnerable to those key indicators could promote investment in power outage risk mitigation in hurricane-prone areas.

In short, the present study has revealed that sustainable management of lifecycle costs, especially in the operation and management stages, can be enhanced by recognizing that power outages during hurricanes have an inevitable relationship with natural hazard indicators. It has also demonstrated that state-wide power outage data from past hurricanes can be used to identify loss correlations for future ones.

## 6. Conclusions

Due to its lengthy coastline and geographical placement, Florida has suffered disproportionate financial losses and infrastructural damage from hurricanes as compared to other U.S. states. Hence, it is a site of great demand for hurricane damage prediction models. The results of the present multiple-regression investigation of the relationship between natural hazard factors and power outages during Hurricane Irma in 2017 can help us better understand electrical power systems' vulnerability to hurricane damage in lifecycle-cost terms, despite the limited sizes of the study area and data sample, and the relatively small number of natural hazard variables that were used.

An integrated consideration of social, economic, and environmental impacts should be required to improve lifecycle cost as part of the broader goal of long-term sustainability. As this research did not have an opportunity to consider a full range of geographical, climatological, and economic factors, future research devoted to advanced lifecycle assessment incorporating natural hazards should give due consideration to such factors. Furthermore, such future research should take account of geographically weighted factors such as coastline length and the spatial distribution of damage, and collect data from a wider array of weather centers. This is because the relationship between power outages and natural hazards could differ considerably across geographical locations and their local weather patterns. Other natural hazard indicators such as wind-speed motion and movement direction, along with earthquakes and other environmental factors such as proximity to waterways, should also be considered when expanding this study's model to analysis of the long-term sustainability of buildings and other infrastructure in a wide variety of locations and levels of hazard exposure.

Lastly, a vulnerability function can be estimated if the approximate damage costs arising from power outages can be provided as a function of wind speed or rainfall. Such estimation can help with advance prediction of losses caused by hurricanes, according to the intensity of the natural hazard indicators. In the future, if electric power systems' initial times of failure and restoration time become available, an integrated response and restoration strategy can also be developed to minimize losses to such systems in hurricane prone-areas.

**Author Contributions:** Conceptualization, S.-G.Y.; Data curation, S.-G.Y., S.S., and J.-M.K.; Funding acquisition, J.-M.K.; Investigation, S.-G.Y. and K.S.; Methodology, S.-G.Y. and J.-M.K.; Project administration, J.-M.K.; Software, J.-M.K.; Supervision, J.-M.K.; Validation, S.-G.Y. and J.-M.K.; Resources, J.-M.K.; Writing—original draft, J.-M.K.; Writing—review and editing, S.-G.Y. and K.S. All authors have read and agreed to the final version of the manuscript.

**Funding:** This research was funded by the Basic Science Research Program of the National Research Foundation of Korea (NRF-2020R1F1A1048304).

**Conflicts of Interest:** The authors declare no conflict of interest.

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
