# Peer review of "Identifying Risk Indicators for Natural Hazard-Related Power Outages as a Component of Risk Assessment: An Analysis Using Power Outage Data from Hurricane Irma"

_sustainability, doi:10.3390/su12187702_

Round 1

Reviewer 1 Report

  1. Lines 2-4: In my opinion the content does not fully match to its title. If Authors mention about a lifecycle assessment, Reader expects more and directly described relation of the results to the lifecycle assessment. This is missing in the paper. But the research is very valuable even without the lifecycle assessment connections. So, I recommend Authors to think about changing the title to be more suitable for the content.

  1. Lines 17-32: In Abstract, Authors pay a main attention to the research background (more than a half of the entire content). Respecting ‘a marketing’ character of Abstract is strongly required. Thus, I recommend to reformulate the text to give general information about results and more deeply formulated practical value of them (why the results are useful to the construction and insurance industries and “(…) policymakers tasked with mitigating the effects of natural disasters and 31 building more sustainable societies and economies”).

  1. Line 39: Following the information source [1], the sentence “(…) natural hazards such as hurricanes have been occurring more frequently because of global warming” is proper to the recent decades, counting till 2007. This is why I recommend to use more actual information source to justify the assumption (e.g. UNDRR reports).

  1. Lines 57-58: Authors say that “(…) findings can usefully inform emergency-response and restoration plans for dealing with these extreme events” with no related information in further part of the paper. I recommend an additional effort to make Introduction chapter and Discussion chapter more coherent and merit-connected.

  1. Line 86: As ROF is defined here, there is no reason why it should be additionally defined in lines 226, 236, 286 and 289.

  1. Lines 238-259: A subchapter titled ‘Multiple regression analysis’ seems to describe a methodological layer of the research, not results. Consequently, I recommend to move this content to the ‘Research methods’ chapter, add information reflecting lines 266-272 and lines 277-283 (not move the content but add information introducing to it in the ‘Research methods’ chapter – to clear the paper structure).

  1. Lines 284-293: The text seems to describe a methodological layer of the research, not the results. I recommend to move this section to the ‘Research methods’ chapter.

  1. In general, sustainability issues in the manuscript are considered very indirectly. I recommend to describe a simple, direct thinking line, connecting the research with the sustainability state-of-the-art and development. Additional references to the Sustainability Journal could be helpful, for sure.

  1. In general, Authors use complex sentences to express their thoughts. As far as I am concerned, this could be challenging for Readers to fully and relatively quickly understand an entire message. Thus, I recommend to cut complex sentences to the shorter ones.

Reviewer 2 Report

The statistical correlation analysis presented is interesting as a first step in the resilience analysis. The analysis should be deepened from a cost perspective to provide a more comprehensive assessment from an application perspective.

Round 2

Reviewer 1 Report

All my remarks have been taken into consideration. The manuscript has been evaluated to increase the scientific soundness and overall research presentation quality.

Reviewer 2 Report

In the reviwer opinion the new version is fine.

This manuscript is a resubmission of an earlier submission. The following is a list of the peer review reports and author responses from that submission.